# IoT Solution for Smart Cities’ Pollution Monitoring and the Security Challenges

**DOI:** 10.3390/s19153401

**Published:** 2019-08-02

**Authors:** Cristian Toma, Andrei Alexandru, Marius Popa, Alin Zamfiroiu

**Affiliations:** Department of Economic Informatics and Cybernetics, Faculty of Cybernetics, Statistics and Economic Informatics, Bucharest University of Economic Studies, 010552 Bucharest, Romania

**Keywords:** Internet of Things, IoT edge device security, cybersecurity, sensors, IoT communications protocols, smart-cities

## Abstract

Air pollution is a major factor in global heating and an increasing focus is centered on solving this problem. Urban communities take advantage of Information Technology (IT) and communications technologies in order to improve the control of environmental emissions and sound pollution. The aim is to mitigate health threatening risks and to raise awareness in relation to the effects of air pollution exposure. This paper investigates the key issues of a real-time pollution monitoring system, including the sensors, Internet of Things (IoT) communication protocols, and acquisition and transmission of data through communication channels, as well as data security and consistency. Security is a major focus in the proposed IoT solution. All other components of the system revolve around security. The bill of the materials and communications protocols necessary for the designing, development, and deployment of the IoT solution are part of this paper, as well as the security challenges. The paper’s proof of concept (PoC) addresses IoT security challenges within the communication channels between IoT gateways and the cloud infrastructure where data are transmitted to. The security implementations adhere to existing guidelines, best practices, and standards, ensuring a reliable and robust solution. In addition, the solution is able to interpret and analyze the collected data by using predictive analytics to create pollution maps. Those maps are used to implement real-time countermeasures, such as traffic diversion in a major city, to reduce concentrations of air pollutants by using existing data collected over a year. Once integrated with traffic management systems—cameras monitoring and traffic lights—this solution would reduce vehicle pollution by dynamically offering alternate routes or even enforcing re-routing when pollution thresholds are reached.

## 1. Introduction

The air pollution in major cities may reduce life expectancy by up to 22 months. However, according to a study that was made by the World Health Organization (WHO) in May 2016 and updated in 2018, most big cities are facing this issue. More than 80% of urban areas reach air pollution levels well above acceptable values [1,2]; “around seven million people die every year from exposure to fine particles in polluted air that lead to diseases such as stroke, heart disease, lung cancer … and respiratory infections”. The World Health Organization provides an interactive map based on estimate models, which considers fine particulate matter (PM2.5) in micrograms/cube meter.

Therefore, this has become a growing problem in recent years and its effects are visible in everyday life.

The data obtained by monitoring the busiest urban areas prove that pollution impacts not only our own lives but also the lives of the generations to come. Therefore, it is very important to try as much as possible to be careful and control the harmful emissions that we release into the atmosphere.

The Internet of Things (IoT) solution aims to monitor pollution levels, especially levels present within cities, by using several IoT devices and attached sensors. The idea involves having multiple stations placed in several areas of cities. These stations periodically upload and send data to the IoT cloud. For the proof of the concept and prototyping, the authors used development boards, such as Nitrogen iMX 6 or Raspberry Pi, but for the final IoT solution, properly calibrated high-quality sensors and industrial IoT gateway devices, such as HMS Netbiter or equivalent equipment, are targeted. The key differentiator between this solution and other solutions is the designed security for the IoT gateway and communications to the IoT Clouds. The artificial intelligence (AI) and ontologies and other technologies may be used for better prediction and data analysis.

One purpose of the paper is to collect, analyze, and process the data received from sensors in order to obtain real-time monitoring of the air quality, as well as determine the security and consistency of the collected data. Therefore, it is necessary to design and develop new types of public services, such as:Determination of areas less polluted at a certain moment in time—parks, markets, or any other public area.Determination of the most polluted areas, in order to avoid them.Triggering specific public actions, such as partial road closures, in order to decrease the current pollution level or the triggering of alarms if excessive measurements have been recorded.Identifying the least polluted area in real estate.

A city is “smart” if it uses different types of sensors and devices to collect data and provide information that is used to efficiently manage resources. Data can be processed and analyzed in order to monitor and manage the traffic and transport systems, waste recycling, water supply networks, and other public services.

Stations, located in crowded areas of the city, periodically read a number of samples containing data from attached sensors (Figure 1). At a set time, the station processes an average sample to reduce erroneous data and sends it to the cloud platform. The data is transferred using a low-resource protocol.

The main contributions of the authors in this paper are the following:Design of a large scalable architecture and infrastructure for pollution metric collection within an IoT smart city solution.Design and implementation of the proof of concept (PoC) of an IoT architecture to collect pollution metrics within a smart city using the best practices in software development technologies.Analysis of the technologies and platforms that may be used within a pollution monitoring solution.Security of the Message Queuing Telemetry Transport (MQTT) communication protocol implementation for the collected data, in order to avoid indirect attacks on the data collection process. The system also needs to be resilient to malicious threats (e.g., cyber terrorism) that could represent high risks once the integration with any traffic control system is made.

The paper is structured in five sections, starting from the sensors’ data collections, IoT gateway, communications protocols, and IoT cloud to the security challenges and conclusions. The second section presents the existing solutions for pollution monitoring and shows the differences between them.

The third section shows the bill of the materials and the communications protocols necessary for designing, developing, and deploying the IoT solution. It also emphasizes the architecture of the components, the data flow implementation details, and the edge connectivity to the IoT cloud.

The fourth section presents the security challenges and the PoC approach merits. The last section shows conclusions and future work, as well as the results of the pollution monitoring gathered for one year inside a major city (e.g., Bucharest, Romania, European Union in 2018).

## 2. Related Works

According to [3], air quality assurance is a very big concern for humanity, especially for people living in urban areas. A custom extensible Air Quality Internet of Things (AQIoT) device platform is capable of making measurements and stores all collected data in the cloud. The components used in the first version of the device cost around US$900 dollars and are based on a Raspberry Pi3 Model B. This version was tested with six devices in Southampton in the UK. Communication between the devices is achieved via an Long Range Wide Area Network (LoRaWAN). The second version of the device—costing around US$1000 dollars—was developed next. Another approach is presented in [4], using a Raspberry Pi2 Model B board. Other perspectives on solving similar problems are presented in [5,6].

LoRaWAN is also used in a Romanian solution named uRADMonitor [7]. It equips smart city projects with sensors for various pollutants. This solution was used in Alba Iulia, with 15 sensors installed on 15 buses. This novelty of this approach is in the way the sensors work and the platform’s ability to render maps using the measurements’ decks and dashboards. Samples of the collected data can be accessed at [8].

Another pragmatic project is RadonAir, which was developed by the Dositracker start-up. It aims to reduce the concentration of radon in dwellings and buildings [9]. Another research that analyzed smart buildings is [10], in which the authors proposed a system that focuses on efficient energy use for data collection.

According to [11], as people spend 90% of their time indoors, home air quality is also very important. The iAir system is used for collecting data about ammonia, carbon monoxide, nitrogen dioxide, propane, butane, methane, hydrogen, and ethanol. The iAir provides real-time alerts when the concentration of one of these gazes is excessive. The system uses an ESP8266 as a microcontroller and an MICS-6814 sensor for measurements. The collected data is stored in ThingSpeak, which allows data analysis on the time series. The micro computing unit (MCU ESP8266) is also part of another solution presented in [12].

Internet for Things technology is used for the monitoring of both air and water pollution. Another IoT project with (Supervisory Control And Data Acquisition (SCADA) technology for water quality monitoring is presented in [13]. The security of these systems is an important aspect and several studies, such as [14], focus on its implementation in data integration and air quality monitoring.

The second version of the solution aims to use the IoT-NB 5G GSM (NarowBand IoT) network in a hybrid approach with Wi-Fi, SigFox, and LoRaWAN and to enhance security by using the Java Card secure element in within an industrial IoT gateway. This uses at least an IP of 67 because of the outdoor exposure of the station. Development boards, such as Raspberry Pi 3 and Nitrogen iMX6, are suitable candidates for proof of the concepts and fast prototyping, but in real solutions they are not reliable.

## 3. IoT PLATFORM for Monitoring Smart Cities’ Pollution (IoTP4mSCp Solution) Architecture, Implementation, Material, and Methods

The Internet of Things platform for monitoring smart cities’ pollution (IoTP4mSCp) contains the following components:■Wired and wireless sensors for various metrics measurements;■IoT gateway(s)/node(s) for data collection—for the moment, development boards are used for the proof of the concept;■IoT communication middleware with security features for sending data to IoT clouds.■IoT cloud solution for mathematical, statistics, and artificial intelligence (AI) models for data analytics and data science techniques; the authors are still working on these models.

The following subsections present details of the implementation. The first focuses on the components’ configuration—cloud services, MQTT broker, Node-RED, and MongoDB database as well as on the implementation of the platform solution through the gateway application.

### 3.1. IoTP4mSCp Architecture and Data-Flow

The data flow diagram (shown in Figure 2) presents the logical flow of data processing and transmission functions through the current architecture.

The architecture (Figure 2) has the following components:IoT Edge Devices
○IoT wired (Inter Integrated Circuit (I2C), Serial Peripheral Interface (SPI), and Universal Asynchronous Receiver/Transmitter (UART)) and wireless sensors (ZigBee/Z-Wave)used to collect the following data using 5 s sample rates:■Temperature (°C), pressure (hPa), altitude (m), relative humidity level (%), carbon dioxide (CO_2_ ppm), carbon monoxide (CO ppm), ammonium (NH_4_ ppm), methane (CH_4_ pbm), detected Wi-Fi networks, and their signal strength in decibels.○IoT Smart Objects■IoT nodes—for the prototype and proof of concepts, it is ok to use ESP8266, but in production industrial/outdoor MCUs (micro computing units) should be used.■IoT gateways—for the proof of concepts, development boards (e.g., Raspberry Pi 3 and Nitrogen iMX6) are adequate, while in production, real IoT gateways, such as HMS Netbiter or Eurotech or equivalents, should be used. These are a bit more expensive than the IoT nodes but have greater computation power and their role is to supervise the IoT edge devices’ communications and cooperation.IoT networks/internet—communications to the Internet/IoT cloud are important and need to be reliable and secure, ensuring at a minimum coverage across the city. The following network communications may be used: GSM (2G/3G/4G-LTE/5G—IoT-NB), SigFox, LoRaWAN, and Wi-Fi.Solution clouds—the prototype used two types of clouds:○IaaS/CaaS (Amazon EC2 with containers as Docker and Kubernetes)—a virtual machine with a Docker/Kubernetes container, where the authors deployed: The webserver (Apache Tomcat 9.0) with RESTful services, the MQTT broker (Mosquitto), node.js 9+ and Node-RED frameworks, and the non Structured Query Language (NoSQL) database (MongoDB).○IoT PaaS-dedicated cloud (such as Google IoT, Amazon Web Services IoT, IBM Bluemix IoT, and Oracle IoT) is used to not be dependent on any cloud provider. These clouds are used for dedicated IoT time series analytics. The data processed in this section is transferred to either the enterprise systems for business intelligence, reports, and forms, or to the front-end consumers that allow data management and reporting.Enterprise Systems
○business intelligence (BI), artificial intelligence (AI) and data analytics clouds or enterprise systems may be used to provide a strong interpretation of the process data. This part is in progress within the research project of the authors.Front-end devices for management and reporting (e.g., smart city pollution map creation).

IoT gateway pollution monitoring stations are MQTT-subscribed to the broker topics: “/stations” and “/stations/ {identifier station}”, where {station_identifier} represents the Media Access Control (MAC) address of the station network interface card if Wi-Fi/Ethernet/GSM networks are used. In these topics, they are waiting for configuration messages. After initialization, the stations collect data from the sensors at a specific interval and publish an MQTT message to the Mosquitto broker hosted on the AWS EC2 side.

The Node-RED component persists the data through the MQTT protocol. It launches a stream that listens to messages posted on the topic/sessions, takes over their payload, validates it, and then stores it in MongoDB.

The IoTP4mSCp cloud/backend component provides RESTful services for interacting with the MongoDB database (CRUD operations) and with the MQTT broker. The messages, which are published in the above broker topics, modify the configuration parameters of a station—enable/disable, interval, where data is retrieved/sent.

### 3.2. Hardware Bill of Materials

To monitor the level of air pollution and air quality, a development board (such as Raspberry Pi or Nitrogen iMX) station records the following data with attached sensors: Temperature (°C), atmospheric pressure (hPa), altitude (m), humidity level (%), carbon dioxide level (CO_2_ ppm), carbon monoxide (CO ppm), methane (CH_4_ pbm), and ammonium (NH_4_ ppm). Also, data on WI-FI networks identified in the vicinity of the station and the level of their signal are recorded. The components that make up a station are shown in Table 1.

The development board (for the proof of concept station IoT Gateway) Raspberry Pi 3 Model B as an alternative to the Nitrogen iMX board is a single board computer (SBC) of a credit card size with a BCM2837 Quad Core (4x ARM Cortex-A53, 1.2 GHz), 1 GB RAM, Wi-Fi, Bluetooth 4.1, and Bluetooth low energy (BLE). Raspberry Pi has no internal flash memory and it uses a micro SD card for storage. The model also features four Universal Serial Bus (USB) ports, High Definition Multimedia Interface (HDMI) port, and 40 General Purpose Input/Output (GPIO) pins. The Raspberry Pi board is powered by a micro USB port that supports a 5 V input voltage. Raspberry Pi 3B contains a GPIO 40 pin layout—digital inputs that can connect a lot of sensors/modules. There are two 5 V pins, two 3.3 V pins, and an eight-pin ground.

Raspbian is the official operating system released by Pi Foundation for the first time in June 2012. It is based on Debian and is optimized for Raspberry Pi ARM processors. The authors used several distributions (Jessie and Stretch) and Java 8 SE-e (standard edition embedded edition), Python 2.7, and 3+ and node.js 8+ with Node-RED packages for the sensors’ data gathering.

All 40 GPIO pins on the Raspberry Pi card support digital inputs, so an analog-to-digital converter is needed to connect the analog sensors and read the values they provide.

The MCP 3008 is an analog-to-digital converter, with which an analogue size can be accepted at the input, and a number representing an approximation of the analog signal of the input signal is outputted. This model works with eight analogue channels, allowing up to eight analog sensors to be connected. The working voltage is 3.3 V and the operating temperature is between −40 and 85 °C.

The SNS-MQ135 is an electrochemical sensor that measures air quality and is capable of detecting levels of carbon dioxide (CO_2_), ammonium (NH_4_), ethanol (C_2_H_6_O), and toluene (C_7_H_8_). The sensor has an analogue and digital output, uses a working voltage of 5 V, and has a current consumption of ~40 mA. The working temperature is between −20 and 50 °C.

MQ9 is a gas sensor that measures the level of carbon monoxide (CO) and methane (CH4). Like SNS-MQ135, this sensor has analogue and digital output, a working voltage of 5 V, and a current consumption of ~40 mA. The working temperature is between −20 and 50 °C.

The two sensors in the MQ series (MQ135, MQ9) require a pre-heating period of at least 24 h, when commissioned for calibration.

SYH-2R is an analog sensor for the detection of relative humidity levels. The working voltage is 5 V and the operating temperature is between −20 and 85 °C.

BMP280 is a sensor designed by Bosch to detect temperature (±1 °C accuracy) and barometric pressure (±1 hPa accuracy). It contains a voltage regulator and can use a working voltage of 3.3 to 5 V. The operating temperature is between −40 and 85 °C. This sensor can communicate through both the Serial Peripheral Interface (SPI) interface and Inter-Integrated Circuit (I2C) bus.

### 3.3. Hardware Diagrams

Fritzing is an open-source software product used to ease the connection of electronic devices and circuits. It allows the design of circuit diagrams and diagrams, as well as the design of Printed Circuit Board (PCBs). The schema and connection diagram were designed by Fritzing. When assembling the components, the mother-to-male connection wires and the breadboard were used to extend the 5 V and ground lines (Figure 3).

The analogue sensors are connected to the MCP3008 converter as follows: The SNS-MQ135 is connected to the analogue channel, A0; the MQ9 is connected to the analogue channel, A1; and the humidity sensor, SYH-2R, is connected to the A2 channel. The converter connects through the SPI interface to the Raspberry Pi board (Figure 4).

The connection of the analog-digital converter to Raspberry Pi is done via the SPI interface (Figure 3) as follows: VDD pin connects to 3.3 V—GPIO 01, the GND pin connects to the ground pin—GPIO 06, the SCLK pin connects at SPI_CLK—GPIO 11, the MISO pin connects to SPI_MISO—GPIO 09, the MOSI pin connects to SPI_MOSI—GPIO 10, and the CE pin connects to SPI_CE0—GPIO 08.

Interfacing the BMP280 temperature and pressure sensor with Raspberry Pi is done by the I2C protocol as follows: The SDI pin on the sensor connects to SDA1—GPIO 02, the CS pin connects to SCL1—GPIO 03, the GND pin to the ground-board on the breadboard, and the pin 3.3 V to the power plug on the breadboard.

Connecting the status LEDs is done through the following pins: Green LED connects to GPIO 21, blue LED connects to GPIO 12, and red LED connects to GPIO 05. All LEDs connect to the line ground on the breadboard.

### 3.4. Sensors’ Communication Protocols and Reading Data from the Sensors

By using a communication protocol, two or more entities in a communication system exchange information. The interface of the analog sensors (SNS-MQ135, MQ9, SYH-2R) and digital sensor (BMP280) with the IoT gateway development boards can be achieved through the following communication protocols:**Wired sensors:**○**SPI (serial peripheral interface)**—Serial peripheral interfa is a synchronous communication protocol used to transmit information between master devices (such as IoT gateway/node development board) and peripheral devices. The SPI bus can operate with a single master device and one or more slave devices. In the authors’ solution, the analog-digital converter MCP3008 is connected to the IoT nodes/gateways development board by SPI, so the latter has the role of the master and the Analog-to-digital Converter (ADC) is the slave. SPI uses four logical signals to communicate with the peripheral device. These are: Serial clock (SCLK), master input slave output (MISO), master output slave input (MOSI), and slave select (SS). The clock is configured by the master at a supported frequency less than or equal to the maximum frequency supported by the slave and then the master configures the slave on the “low” level for the selected chip. At each clock cycle, the data is transmitted in full duplex mode as follows:■The master sends one bit to master-output/slave-input and the slave reads it.■The slave sends one bit to master-input/slave-output and the master reads it.○I2C—The inter-integrated circuit (I2C) communication protocol was developed by Philips Semiconductor in 1980. It allows the communication of several slaves with one or more master devices. Similar to the SPI interface, the inter-integrated circuit is designed for small data transfer, but unlike SPI, the I2C protocol only needs two signal wires to perform data interchange. Each I2C bus is made up of two signal lines: Serial clock line (SCL) and serial data line (SDA). The clock signal is generated by the master, but some slave devices can force the clock to a low level to delay the master sending more data (or require more time to prepare the data before the master takes over).Wireless sensors—these sensors will be added in the next version of the solution:○BLE—Bluetooth low energy will be used for direct connection with sensors that are linked to IoT nodes instead of directly in the IoT gateway.○Z*-protocols—these will be used to decrease the number of IoT nodes and gateways and to increase the number of the smart sensors and also to optimize the power consumption on the edge devices (ZigBee and Z-Wave are more efficient in terms of power consumption that Wi-Fi).■ZigBee—the published version is now Zigbee 3.0—the complete IoT solution communications, from the universal language that allows smart objects to work together to mesh networks. (Z-Wave protocol is used mainly with in-home automation and it is an alternative for ZigBee).

Data collection by the attached sensors is achieved both through the SPI interface (for communicating with analog sensors) and via the I2C interface (for communicating with digital sensors). To read data from sensors via I2C, the authors used the Java Device Input/Output (DIO) library (https://openjdk.java.net/projects/dio/). Bosch, the manufacturer of the BMP280 sensor, offers a Java implementation to retrieve the data. To read data from sensors via SPI, we needed a way to interact with the MCP3008 analogue-to-digital converter. Therefore, the authors created an AnalogDigitalConverter class (Figure 5) that implements the Singleton design pattern and uses the Java jdk.dio (Device I/O) library (used in both Java SE-e standard edition embedded and Java ME—MicroEdition) to communicate with the GPIO pins. Next, the authors created an AnalogSensor abstract class that implements the AnalogSensorOperation interface and contains an AnalogDigitalConverter, the analogue channel where the data is read and the gross value is returned by the sensor.

Thus, all analog sensors will inherit the AnalogSensor class and implement the interface method to provide processing of the readable gross value.

### 3.5. IoT Cloud and Back-End Services

The application from the IoT gateway communicates recorded data using the MQTT protocol with its implementation from Eclipse Foundation—Mosquitto MQTT. This implies the existence of a broker that has the role of selecting and filtering incoming messages. IoTP4mSCp uses cloud services, such as Amazon EC2 IaaS (infrastructure as a service) or Google containers as a service cloud to host the MQTT broker and the polluting station management/monitoring web application. If the authors use Amazon web services (AWS)/Google/Oracle IoT Cloud(s) or PaaS (platform as a service), then the application will directly access the features provided by these IoT-dedicated clouds.

#### 3.5.1. IoT IaaS Cloud for Data Storage and Processing

The Amazon elastic compute cloud (EC2) is an infrastructure as a service (IaaS) cloud that offers a secure computing capability in the AWS cloud. EC2 courts can be resized and their number can be scaled according to requirements. These can be launched in multiple locations and geographic regions called availability zones (AZs). Each region is comprised of several AZs in distinct locations, interconnected by low-latency networks.

Amazon EC2 supports multiple operating systems for which the users should pay licensing fees, such as Red Hat Enterprise, SUSE Enterprise, Oracle Enterprise Linux, UNIX, and Windows Server, and free operating systems, such as Amazon Linux, Ubuntu, and CentOS.

In terms of security, the cloud allows users to create security groups that behave like a virtual firewall that controls inbound and outbound traffic. Thus, specific ports for specific IP addresses can be restricted.

Internal storage of an instance data is done using an AWS (Amazon web services) elastic block Store (EBS) instance. It provides data storage volumes and is automatically replicated for each availability zone (AZ) to provide high availability.

To create the EC2 instance, an Amazon machine image (AMI) must be selected that contains the software configuration (operating system, applications, preinstalled utilities). We chose Amazon Linux AMI 2018.03.0 (Figure 6), which is a RHEL 5.x-based distribution and includes AWS command line tools, Java 1.8, Python, Ruby, and Perl. For the instance type, we chose t2.micro with 1v CPU of 2.5 GHz and 1 GB of Random Access Memory (RAM). The storage is provided by an 8 GB elastic block store volume.

For the full launch of EC2, we needed to define a security group. One instance can be associated to one or more security groups. The security group used (called RDS-Rpi—Figure 7) for this instance is configured to allow inbound ports 22—for SSH—Secure Shell, 8080—to access the web server of the solution and 1883—for the Mosquitto MQTT broker.

After launching the EC2 instance (Figure 8), a private Privacy Enhanced Email (PEM) key “aws-pi.pem” (PEM—privacy enhanced mail file is a X509 digital certificate encoded in Base64) is downloaded to access the instance using the SSH protocol. Authentication is done with the user name “ec2-user”.

Figure 8 shows the public IP of the back-end instance and the public DNS (domain name service) if it is necessary to be accessed from the Internet.

#### 3.5.2. Data Persistence and Server-Side NoSQL Database Configuration

The MongoDB database is hosted by the mLab platform through Amazon web services. mLab offers a sandbox with a 500 MB MongoDB instance. Adding a database user is done from the mLab platform through the add database user form (Figure 9):

Storage of collected data from the IoTP4mSCp IoT gateway is performed using a non-only SQL (NoSQL) MongoDB cloud database. Using a NoSQL database under this project is ideal for storing samples recorded by stations and processing them to be distributed through MapReduce.

MongoDB is a non-relational, open-source database developed in C++ and released publicly in 2009. MongoDB is document-oriented, delivering high performance and scalability. Unlike a relational database, where data is stored in tables, in MongoDB, data is stored as JavaScript Object Notation (JSON) documents in collections. These collections do not impose a certain scheme, so the documents have a dynamic scheme and can have a different structure.

The persistence of the data recorded by the stations in the MongoDB database is carried out through two collections: Stations—in which data is stored on the stations (their name, unique identifier, geographical position), as shown in Figure 10a, and probes—a collection in which the data persistence is recorded by an IoT gateway development board station (air quality level, detected Wi-Fi networks, and their signal)—Figure 10b. Also, the IoTP4mSCp solution has web interfaces, which implies the existence of users who administer the stations. These are stored in the users’ collection in JSON format as in Figure 10c.

The JSON stored into the MQTT broker and then transferred to the MongoDB are next inserted into the distributed relational database for big data processing and reports’ dashboard creation.

The Server Side MQTT Mosquitto Broker configuration involves installation of the broker on the EC2 instance and is done using the yum package manager. All non-specified subscription and publishing requests are restricted by setting the ‘allow_anonymous’ to the ‘false’ value.

#### 3.5.3. Node-RED Framework

Node-RED is an IBM-based, flow-based programming tool for connecting hardware, APIs, and online services. It is built on Node.js and uses its non-blocking and event-driven model, making it ideal for running on limited-cloud or cloud-based devices.

It provides a browser feed/flows editor that runs as default on the 1880 port, which can connect a lot of input, output, storage, and JavaScript functions. The feeds/flows can be imported/exported in JSON format—JavaScript Syntax Object Notation (e.g., {“software”:”Node-RED”, “version”:1}).

Node-RED is a flow-based development tool developed originally by IBM for wiring together hardware devices, APIs, and online services as part of the Internet of Things.

Using a Node-RED instance (this can be deployed anywhere, even on localhost) and a development flow, the messages published via MQTT by the stations can be written to the MongoDB instance.

This flow contains the following nodes (Figure 11):MQTT output node (connected to mosquitto broker from AWS EC2).JavaScript function node (collects and validates the payload from the MQTT message).MongoDb input node (connected to mLab MongoDb instance).Debug node (logs the raw MQTT message).

### 3.6. IoT Communications Protocols to the IoT Cloud

As IoT middleware and communications protocols between the IoT gateways and the IoT Cloud for the solution, MQTT(s) and HTTP(s)—REST-API are used.

MQTT (message queuing telemetry transport) is a publish-subscribe-based communication protocol that runs over the TCP/IP protocols’ stack (Figure 12). Filtering messages is done on a topic basis by the broker, and messages are categorized on topics prior to publication.

In the above TCP/IP communication protocols’ stack (Figure 12), the MQTT [15] is an application layer protocol and it does not provide security by design or by default. Indeed, MQTT has a QoS (quality of service) mechanism but this does not replace the lack of the security.

Developed in 1999 by Andy Stanford-Clark (IBM) and Arlen Nipper (Arcom), it was designed to be an extremely simple and easy protocol designed for devices with limited resources, low bandwidth, high latency, or unstable internet connection. The previous principles prove that this protocol is ideal for the machine-to-machine (M2M) and Internet of Things (IoT) domains, where a small footprint is needed. The MQTT protocol has ports 1883 and 8883 (MQTT over SSL) reserved by Internet Assigned Numbers Authority (IANA).

Since March 2013, MQTT has been standardized at OASIS (Organization for the Advancement of Structured Information Standards) and after the protocol specification was published [15], companies, like Eurotech (formerly known as Arcom), have implemented the protocol in their products. Since 2016, MQTT has become an ISO standard (ISO/IEC 20922).

The publish-subscribe paradigm is a messaging model in which publishers publish messages that can be received by a subset or the entire subscribers’ set.

The mechanism by which subscribers receive only a subset of all published messages is called message filtering. There are two forms of filtering: Topic-based filtering and content-based filtering:■In the topic-based approach, messages are categorized and published to topics (also called logical channels). Subscribers to the system will receive messages that are posted to the topics they are subscribed to. The sender is responsible for defining the message classes subscribers can subscribe to.■In the content-based approach, messages are delivered to subscribers provided that the attributes or content of the message correspond to the constraints defined by them. In this case, the subscriber is responsible for selecting and classifying the messages he receives.

Publishing subscriptions are managed by a broker, which carries out the message filtering. All customers need to establish a connection with the broker. The client who publishes a broker message is called a publisher. The broker filters messages and distributes them to interested customers to receive a specific type of message. Customers who subscribe to the broker to receive messages are called subscribers.

An advantage of the publish-subscribe model is the loose coupling of subscribers. Publishers do not need to know of the existence of subscribers. Their main purpose is to publish a message to a particular topic.

After the broker has been configured, the MQTT client on the IoT edge devices may run the MQTT client applications. When starting the application, an MQTT client is initialized to subscribe to the broker hosted on the Amazon EC2 instance at the following topics: “/stations” and “/stations/{MAC_ADDRESS}”, where MAC_ADDRESS is the MAC address of the network card of the IoT gateway development board (e.g., for proof of concepts, Raspberry Pi or Nitrogen iMX6). Subscribing to these topics is done to change the remote station configuration. All stations available on the platform can be modified by publishing a configuration to the topic “/stations” or only a specific station can be modified by publishing a configuration on “/stations” subtopic defined by the MAC address of the board.

For implementation of the message processing, the authors have defined a MQTTClient class that contains a Mqtt Paho client—by Eclipse—and implements the MqttCallback interface. This interface defines the messageArrived() method (Figure 13), in which the current configuration of the station is updated.

In the above method (Figure 13), the messageArrived() method is called by the Paho MQTT library within the JVM (Java virtual machine) with two parameters: The topic name and the message. The received message is encapsulated within the GSON object and then the station configuration is modified by a new thread within JVM. Any occurrence of an exception is captured by the Java try-catch mechanism.

### 3.7. RESTful Services’ Implementation

RESTful services are implemented through the classes: ProbeController, StationsController, and UsersController. They are annotated with @RestController and @RequestMapping specifying the URI of the resource. Adding a polluting sample (Figure 14) is done through a POST HTTP request to “/api/probes”.

The POST HTTP request must contain a request-body in which the entity is defined in JSON format. If the request does not contain a valid body, the client receives an HTTP 400—a bad request error.

### 3.8. Front-End Implementation

The IoTP4mSCp solution’s front-end component is a single-page application (SPA) built into the Angular 5.2-based framework and the angular material library—used to use predefined components, such as dialog windows, buttons, and tables. The project was created using the command line interface (CLI), which facilitates the generation of components, services, and modules.

The user interface includes the following sections: User area—includes authentication, account creation and profile modification, and station area—fetches information about IoT Gateway stations, adding/configuring/deleting them, and the pollution sample area, where detailed reports are presented based on recorded samples (Figure 15).

In Figure 15, for each device, a list of devices from the IoT solution is shown. Adding and configuring devices is straightforward. The user experience is enhanced with the dashboards, such as in Figure 16 and Figure 17.

The stations component is responsible for interacting with the Raspberry Pi pollution monitoring stations. It contains the following sub-components: StationsList—a component that defines a list of stations that is displayed in a grid (Figure 16), StationsCreate—a component that defines a dialog with a form to create a new station, StationsMap—a component showing the geographic position of each station on the map, and StationsConfig—a component that displays a dialog with the station configuration options.

Adding a station is done through the StationsCreate component (Figure 17), which includes a modal dialog with a stepper component in the angular material library. Adding a station involves three steps: Filling in the station details (name, identifier), filling the station position (latitude, longitude), and confirming the entered data. Upon the addition of a station, a POST HTTP request is made to “/api/stations” to the solution backend.

For the data presentation and analysis, we used the ng2-chart guidelines that are based on the Chart.js library. The data-analysis component injects the probes services service through which REST services are called from the solution’s back-end component to retrieve the pollution samples. These are retrieved according to the user defined parameters: The station for which the display is to be displayed, the parameter list for the analysis, the start date, and the end date.

Figure 18 shows the real-time monitoring of the indicators reported from the IoT gateways distributed within the city.

Since the solution’s frontend component is a single-page application, it is necessary to map the requests from Springboot to Angular. For example, when receiving an HTTP GET request of the resource identified by localhost, 8080/dashboard, the Springboot framework looks for a class controller that can handle that request. Therefore, interpretation of the applications by the P.M.A frontend component is done by redirecting them using the following ViewController class (Figure 19).

In Figure 19, the controller class is responsible for creating objects, which handle the flow of the HTTP requests from the web dashboards, in order to access different REpresentational State Transfer (REST) resources or even HTML webpages.

The first prototype implementation did not take into account security—at least the communications ‘authentication and data confidentiality—encryption. The next section presents the MQTT secure option, because by design, MQTT has no security.

## 4. IoT Security Challenges and the Solution’s Merits

The PoC and the proposed solution of the authors in this paper has several security challenges and areas of enhancements and merits. The first subsection of this section highlights the security challenges while the second subsection pinpoints the merits and extensions of the approach.

### 4.1. MQTT Communication Protocol and the Security Challenges

MQTT does not contain security features built-in to the protocol. The reason is to keep it simple and lightweight. Another reason is related to IoT hardware. Usually, the IoT devices have limited computing power and memory capacity [16]. The hardware resources are insufficient for the support of complex cryptographic. Another IoT security challenge refers to the reliability level of the networks used to maintain the IoT devices.

A security-related implementation exists within version 3.1, where the credentials could be sent through the computer network in an MQTT packet. Encryption of the MQTT packets is handled by SSL/TLS protocol or applications providing encryption of the data. The reserved TCP/IP port for MQTT by using SSL/TLS communication is 8883.

The MQTT documentation states several security challenges [15] that must be handled by the solution providers. Those solution implementations must include mechanisms for authentication, authorization, integrity, and privacy. Guidelines to deploy an MQTT solution consistent with NIST Framework for Improving Critical Infrastructure Cybersecurity are provided by [17]. The NIST Cybersecurity Framework [18] provides specifications and mechanisms for organizations to identify the possible improvements of cybersecurity risk management. In the context of MQTT, this framework is called the MQTT Cybersecurity Framework. The core functions of the MQTT Cybersecurity Framework are used to make assessments of the cybersecurity level for the organization that uses a deployed MQTT solution. The core functions target the following aspects [17]:Identification—establishes what are the systems, assets, data, and capabilities that need to be protected in the context of the MQTT solution exploitation. Also, prioritization and processes to achieve the risk management goals must be established.Protection—develops and implements the right measures to protect or prevent undesirable actions over the MQTT related items identified by the previous function.Detection—develops and implements the right measures for MQTT cybersecurity event occurrences.Response—establishes the appropriate counter measures when a MQTT cybersecurity event occurred.Recover—aims for the activities to restore the organization infrastructure when a MQTT cybersecurity event causes significant damages.

The security levels supported by an MQTT solution are:Network—a trusted connection could be provided by a physically secure network or VPN usage for all communication within the MQTT infrastructure.Transport—encryption ensured by SSL/TLS protocols. Identity of both sides of the communication is proofed by the secure transport protocol implementation.Application—authentication of the devices is ensured by username/password credentials provided by the MQTT protocol. Also, encryption of the payload is possible, but this is constrained by the hardware specifications of the IoT clients.

Even though SSL/TLS usage introduces overheads in network communication and asks for more CPU on the client side, transport-level encryption is critical to protect communication within the MQTT infrastructure. When the transport-level encryption is not feasible due to constrained devices, the alternative approach is to at least encrypt the payload of several packet types [19].

Beside the payload encryption, a stamp attached to the packet PUBLISH could be used. The format of the stamp could be a digital signature, message authentication code, checksum, or hash algorithm.

A different aspect of the SSL/TLS usage is to validate the X.509 certificate on the client side to prevent several types of attacks over the secure connection. The X.509 certificate is provided by the MQTT broker. The recommendation is to use X.509 certificates from trusted CAs.

By using the X.509 client certificate, security improvements are added to the MQTT infrastructure [20], such as:Identity of the MQTT clients is validated.MQTT clients are authenticated at the transport level.Some measures as responses to the MQTT cybersecurity event take place (e.g., lock client before the packet CONNECT is sent).

In addition to SSL/TLS usages, the following security mechanism could be used: Payload encryption and signature, and authorization mechanisms to limit the client access to the MQTT infrastructure resources.

In MQTT, authentication with a username/password is provided by the control packet called CONNECT [15,21,22]. The presence of the connect credentials within the control packet CONNECT is signaled by the connect flags section of the packet, Figure 20.

In Figure 20, for both credentials’ flags, the value 0 means not present. Password flag is dependent on the user name flag being 0 and mandatory if the user name flag is set to 0.

The MQTT username/password credentials are stored in the plain within the payload if the flags are set in the header side of the MQTT packet [23,24]. If the user name is set to 1, then an UTF-8-encoded string is stored within the payload. If the password flag is set to 1, then a field containing up to 65,535 bytes is stored as a password within the payload section. As a result of processing the packet CONNECT with the username/password credentials, the server is able to perform authentication and authorization of the device to access resources within the MQTT-deployed solution.

In addition to username/password credentials, MQTT clients could provide different items to the server in order to perform authentication, like a client identifier or X.509 certificate. On the other hand, the username/password credentials could be used to implement OAuth 2.0. This open protocol is used to allow a secured authorization of the client to the MQTT infrastructure resources based on tokens. An MQTT client is able to connect to a certain broker or to publish or subscribe with an access token. Beside the MQTT protocol specification, additional guidance related to secure MQTT solution implementation could be found on the OWASP platform [25].

### 4.2. Merits of the Proposed Solution

The solution approach has several areas of merits in terms of scalability, reliability, security, and HA (high availability). The merits of the approach presented in this paper are the following:Easy integration of vehicles’ traffic management system (e.g., traffic lights), with the authors’ proposed solution of API as a pluggable component. These APIs are easy to be exposed to any traffic management system in order to create dynamically different routes in order to avoid vehicle congestion.The design of the architecture is suitable for a scalable extension and HA system implementation. The architecture and infrastructure used for pollution indicators’ date collection within an IoT smart city solution presents scalability because it combines with asynchronous message processing (messages-oriented middleware (MoM)/MQTT brokers) with IoT clouds and NoSQL databases for fast data insertions and potential extensions with big data processing from related databases.The deployed IoT gateway software is developed mainly in cross platform software development technologies, e.g., mainly in Java and some optional modules with Node-RED/node.js, and the software is portable and compliant with different development boards/MCUs, e.g., iMX Nitrogen, Raspberry Pi, etc., and industrial IoT gateways, including Eurotech, HMS Netbiter, Dell, etc. The design of the IoT architecture for collecting pollution indicators within a smart city uses the best practices in the software development field.Security of the MQTT communication protocol implementation for the collected data represents a valuable feature, because if the data is altered by a malicious attacker’s vectors from the IoT edge devices to the IoT cloud backend services, then the real-time decisions of traffic management can be dramatically affected and cause massive congestions.For Human-Machine-Interface – HMI related with the mobile, devices fingerprinting is used [26].

The proposed solution can be enhanced in several areas from the data interpretation and systems integrations to cybersecurity. Some of the mentioned ideas are detailed as future work in the next section.

## 5. Conclusions, Data Interpretation and Future Work

After the data is collected from the sensors, the data interpretation process begins. For the moment, the data interpretation is a subject of improvement and needs to be enhanced with multiple features. As an example, we have a few values in Figure 21 showing the pollution sample rates stored from April 2018, and in Figure 22, from December 2018, in a major city—Bucharest, Romania, European Union.

Figure 21 and Figure 22 show partial data collected from IoT gateways deployed within the city and emphasizes according to the table head the following information: The station identification MAC address and the deployment location; the collected data—temperature, humidity, pressure, altitude, CO_2_, NH_4_, CH_4_, CO, toluene, Wi-Fi, and noise decibels; and the date—time of the collected data.

The authors have the full database with the pollution values from different places of Bucharest in 2018 and they considered the WHO database format for reporting directly to World Health Organization. Meanwhile, in the same area and in the same indicators (comparing the carbon-monoxide and carbon dioxide from the same stations in April and in December 2018), it can be observed that the pollution in December 2018 is higher than in April 2018, because of the following:Seasonal influence on the cold weather outside and shopping periods for Christmas, road traffic but also thermo-central consumption of gas for heating households. Also, in 2018, the transit pipes for hot water from the city were more damaged than in 2016 and this cannot be done without investments.Regulations and laws changes—in 2018, it was much easier to bring very old cars—taxes were lower—that produce higher pollution than in 2016 to 2017.The number of powerful Wi-Fi routers increased in 2018 compared with 2017 and also the decibels produced as noise pollution by Wi-Fi access points.Price of gas within gas distributors’ “chains/stations” influenced the traffic.Other factors that may influence the pollution but for which a direct relation cannot be confirmed (covariance between the factor and the pollution level.)

Figure 23 shows, for instance, various areas from Bucharest with the pollution indicators (including the decibels generated by the noise and Wi-Fi networks) in an analytical comparison between 23 April 2018 and 6 December 2018.

The authors enhanced the solution in several areas. Data interpretation and analytics is one of the first focuses as it proves the correlation between different factors. The current interpretation is based on the statistical and business intelligence algorithms, without involving predictive analytics, such as Symbolic Aggregate approximation (SAX) [27,28], or more elaborate Artificial Intelligence/data-mining (AI) algorithms. Also, big data technologies, such as in [29] and Apache Spark platform, may be used as alternative to the SAX in order to produce consistent interpretation regarding the collected data.

The pollution information was summarized in various diagrams/dashboards/maps, such as the one from Figure 23.

The second area of improvement is the IoT edge devices’ replacement with IP67 outdoor resistant devices for all three layers: Sensors, IoT nodes, and IoT gateways. Additionally, the ratio between the number of gateways and sensors needs to be adjusted when more nodes with a relatively low cost but good battery life and communication features appear. The sensors types and calibrations can also be enhanced, as for the PoC, the authors did not use professional sensors from the market.

The cyber-security can be optimized. The authors focused on the security of communications between the IoT gateways and clouds, but there is room for more. By using Java Card 3.1+ technology [30] within different form factors (integrated secure element (iSE), embedded secure element (eSE), etc.), the authors are targeting not only to ensure the security of the end-to-end communications (from sensors to cloud), but also to improve the node/gateway security, by running sensitive and computational cryptographic code on the temper resistant Secure Element with Java Card 3.1+ [30].

The authors also targeted the use of calibrated high-quality sensors and interface implementation for reporting data directly to the World Health Organization, in order to maintain the quality of data for future work.

As future work, the authors will target a case study for applying the PoC with enhancements in different medium cities in cooperation with the local authorities to see the traffic impact simulations and the integration with other smart city systems, because existing projects, such as AirVisual [31], are limited. The AirVisual project is limited only to monitoring the WHO PM2.5 (fine particulate matter) and it does not further analyze harmful particles’ impact on human health. In Figure 24 [31], one may observe the top of the polluted cities in Romania in 2018.

In Figure 24, Bucharest is in fourth place of the most polluted cities from Romania (according with IQAir AirVisual platform), where the average PM2.5 (Particulate Matter less than 2.5 microns in diameter) is 20.3 and the worst months in terms of pollution are January and December, with PM2.5 values of 26.1 and 27.7, respectively. In conclusion, car and transportation traffic is only a part of the problem, because industry development, house/apartments’ HVAC (heat, ventilation, air-conditioning) solutions, waste management, and recycling mechanisms must be improved as well.

The authors focused on the Bucharest traffic congestion as according to the TomTom index [32] in 2018, the city was reported as being in first place, with the biggest value regarding traffic congestion and in to/11 place worldwide after Mumbai, Bogota, Moscow, Istanbul, and other major cities. The congestion level in 2018 according to [32] was 48%, with a 1% improvement on 2017, and with the best day on Sunday, 8 April and the worst day on Wednesday, 5 December. The proposed approach of the authors is “pluggable” in terms of the integration with traffic management systems of smart cities. It can dynamically determine, by making use of drivers’ smart-phone applications and of traffic light management, routes which diminish congestion and help in avoiding polluted areas.

## Figures and Tables

**Figure 1 sensors-19-03401-f001:**
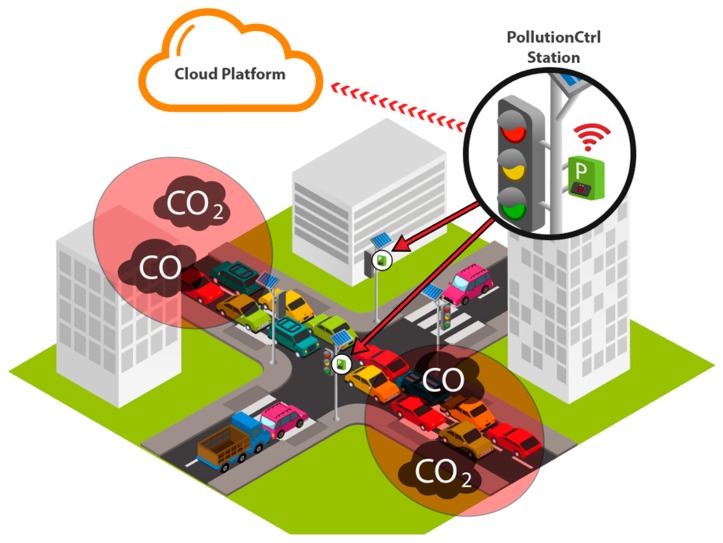
Internet of Things monitoring solution for pollution—generic view.

**Figure 2 sensors-19-03401-f002:**
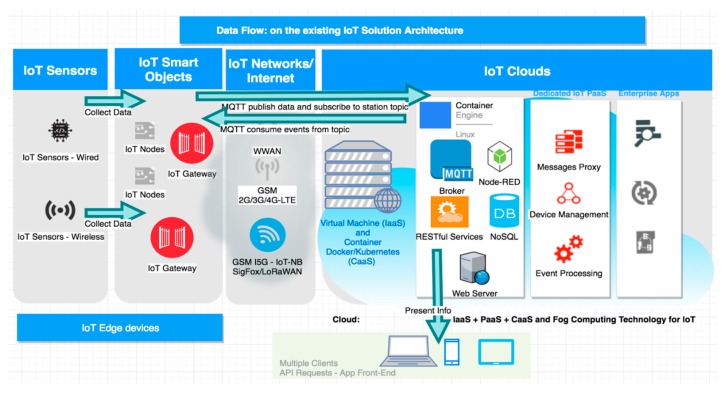
Architecture with data flow.

**Figure 3 sensors-19-03401-f003:**
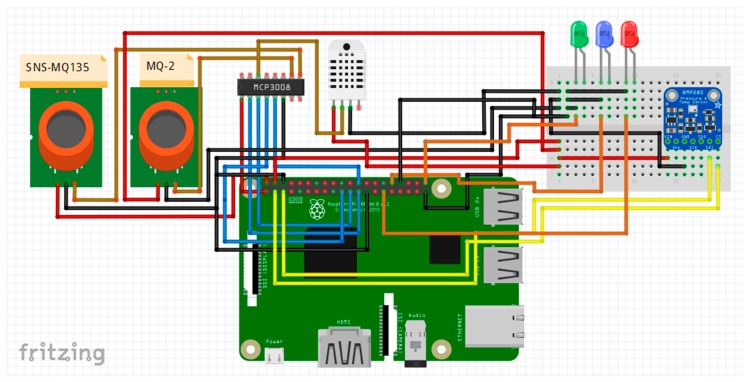
Hardware diagram.

**Figure 4 sensors-19-03401-f004:**
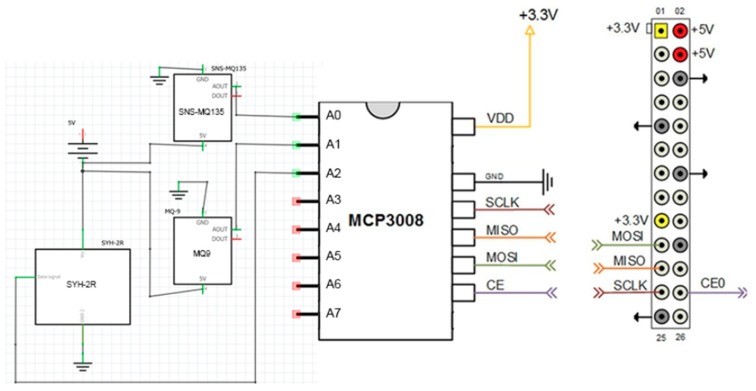
Schematic MCP3008.

**Figure 5 sensors-19-03401-f005:**
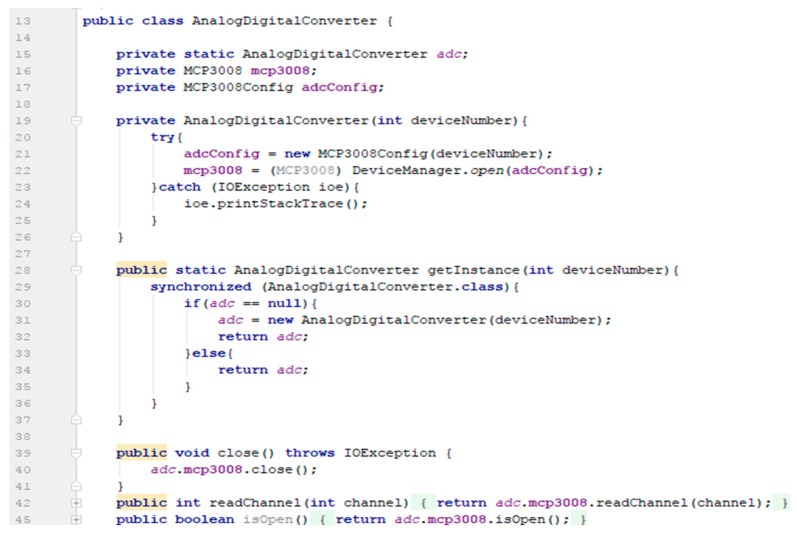
Partial implementation of the class AnalogicDigitalConverter.

**Figure 6 sensors-19-03401-f006:**
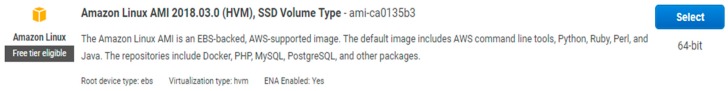
Amazon machine image (AMI) selection.

**Figure 7 sensors-19-03401-f007:**
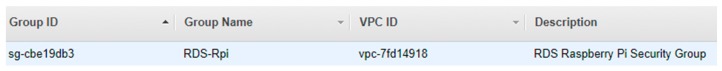
Security group selection.

**Figure 8 sensors-19-03401-f008:**
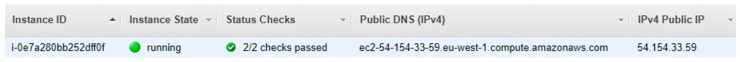
Launching the instance from cloud Amazon Elastic Computing (EC2).

**Figure 9 sensors-19-03401-f009:**
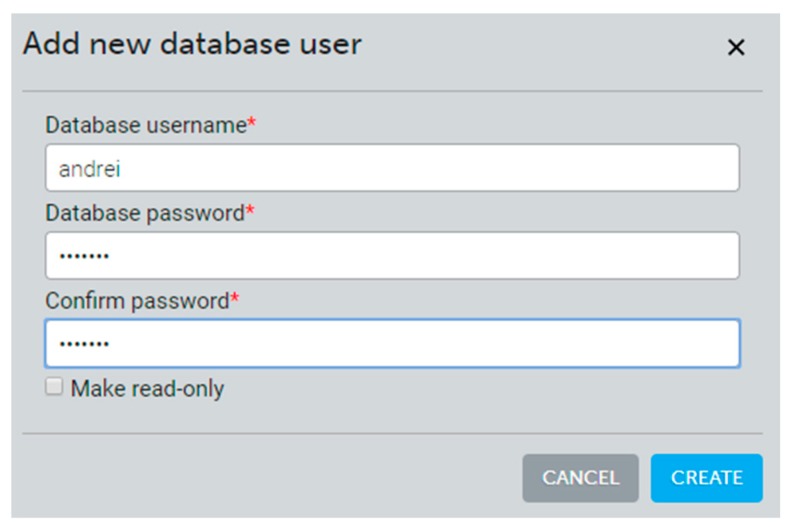
Adding a new user into MongoDB NoSQL Database via Amazon web services’ (AWSs) mLab.

**Figure 10 sensors-19-03401-f010:**
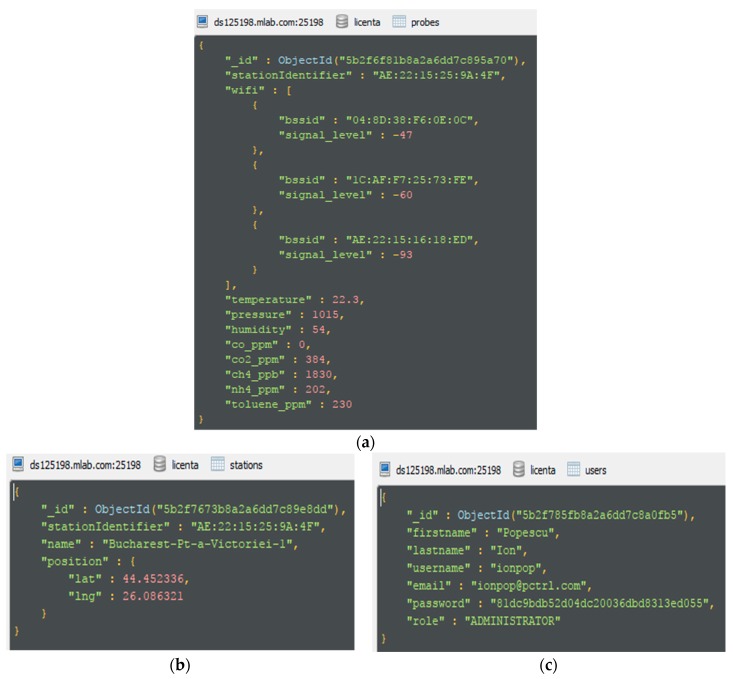
(**a**) The sample rate JSON structure; (**b**) The JSON for the station structure; (**c**) The JSON for the user structure.

**Figure 11 sensors-19-03401-f011:**
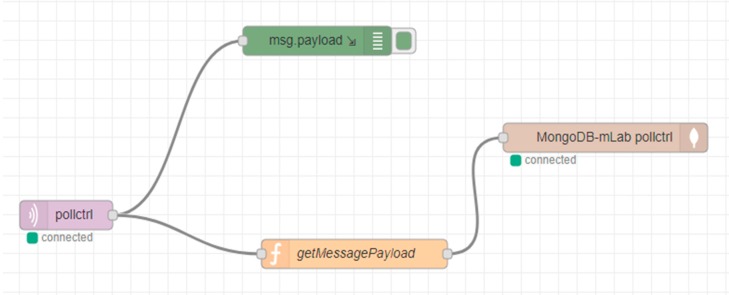
Flow in Node-RED on the server cloud.

**Figure 12 sensors-19-03401-f012:**
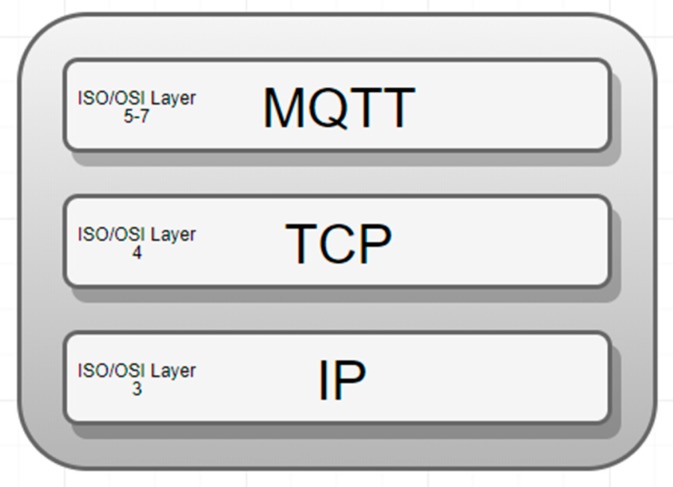
MQTT within TCP/IP communication protocols’ stack.

**Figure 13 sensors-19-03401-f013:**
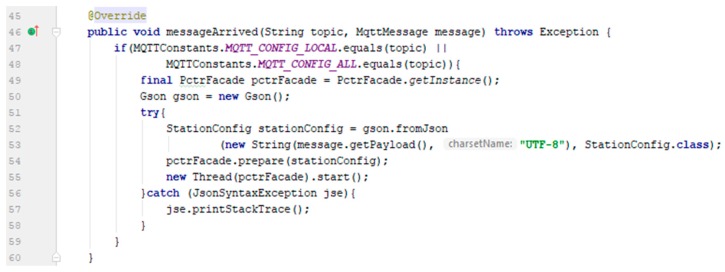
MQTT messageArrived (…) method implementation.

**Figure 14 sensors-19-03401-f014:**
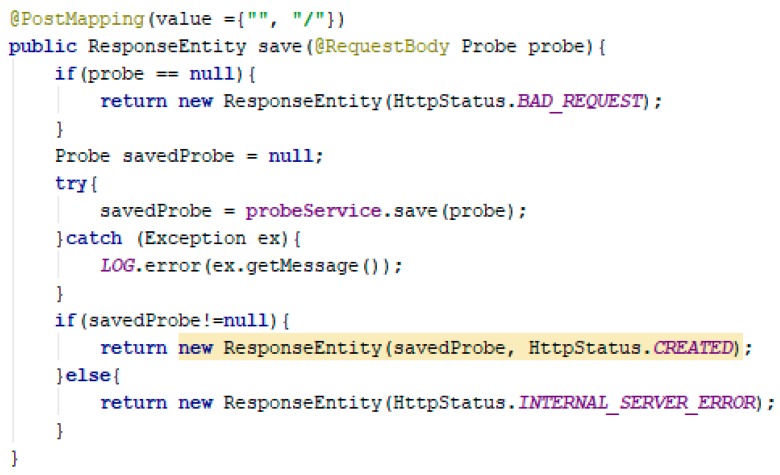
**HTTP** POST probe endpoint.

**Figure 15 sensors-19-03401-f015:**
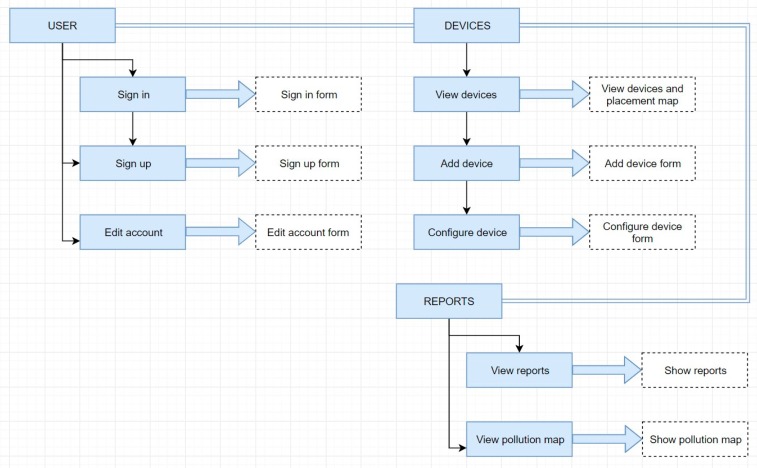
Users’ interface of the front-end component.

**Figure 16 sensors-19-03401-f016:**
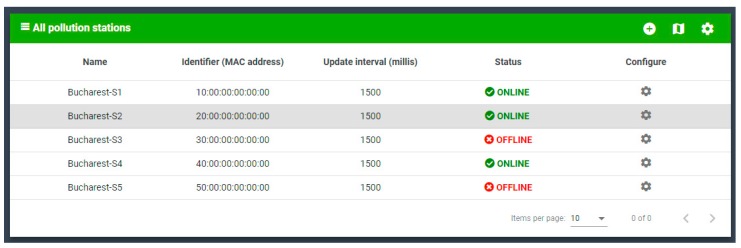
StationsList component.

**Figure 17 sensors-19-03401-f017:**
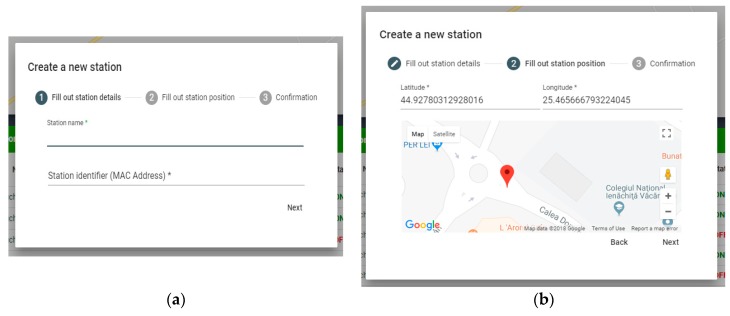
Management operation: add a station. (**a**) Add a new station MAC address; (**b**) Add a new station details.

**Figure 18 sensors-19-03401-f018:**
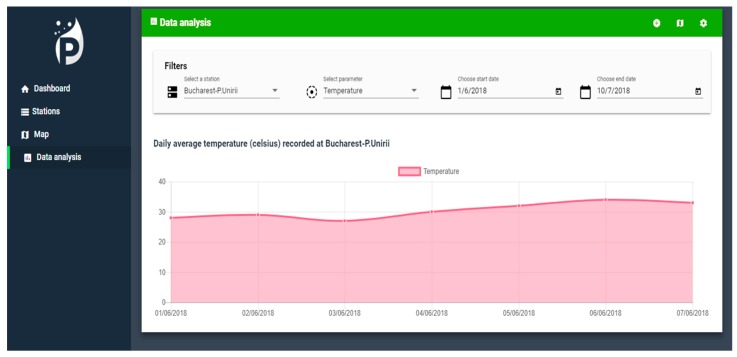
Real time IoT station monitoring.

**Figure 19 sensors-19-03401-f019:**
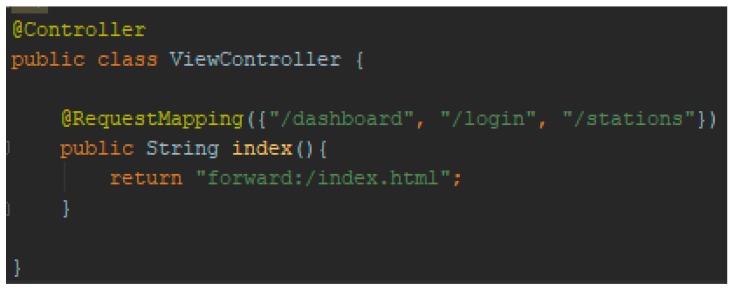
ViewController class.

**Figure 20 sensors-19-03401-f020:**
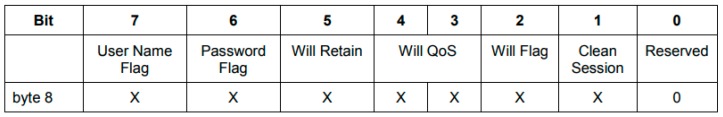
Connect flags section.

**Figure 21 sensors-19-03401-f021:**
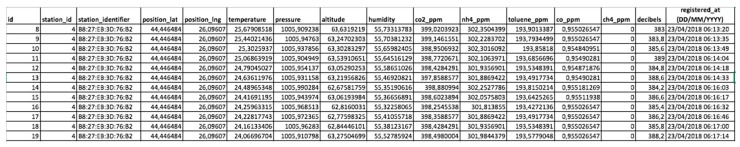
Few (partial) sample data collected from Bucharest on 23 April 2018.

**Figure 22 sensors-19-03401-f022:**
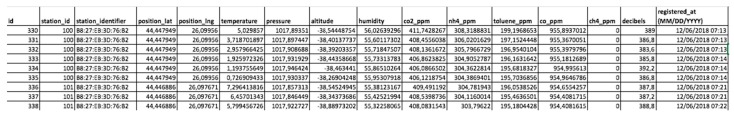
Few (partial) sample data collected from Bucharest on 6 December 2018.

**Figure 23 sensors-19-03401-f023:**
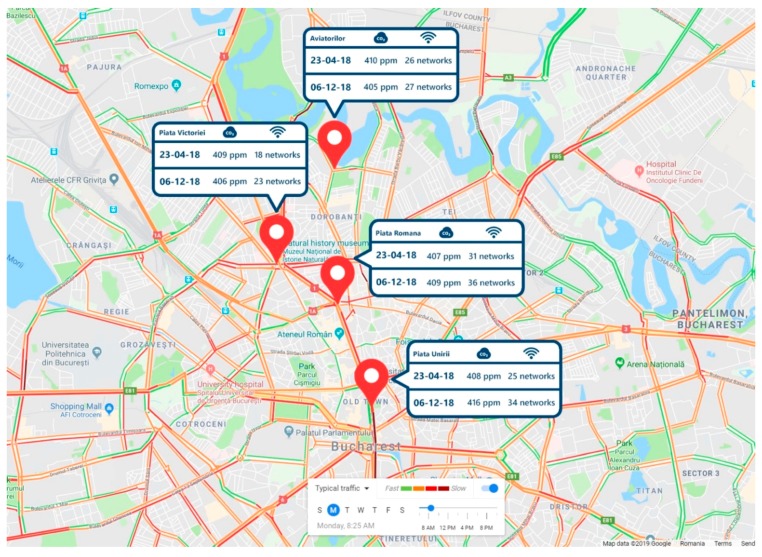
Few (partial) sample data collected from Bucharest on 6 December 2018.

**Figure 24 sensors-19-03401-f024:**
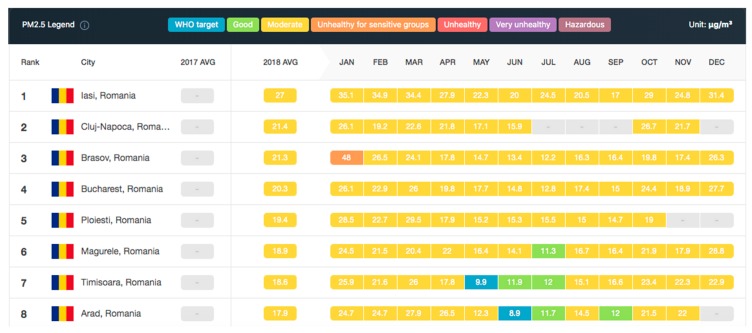
Top polluted cities in Romania in 2018 [31] in terms of PM2.5 according to the AirVisual Project.

**Table 1 sensors-19-03401-t001:** Hardware bill of materials for one station within the IoT solution.

Component	Description	Quantity	URL
**IoT Development board**	The board Raspberry Pi 3 Model B with 40 GPIO pins or Nitrogen iMX board.	1	https://www.sparkfun.com/products/13825
**Power Source**	Power source 2.5 A and 5 V.	1	https://www.adafruit.com/product/1995
**SD CARD 16GB**	SD Card which store Raspbian OS or Embedded Linux Ubuntu.		https://www.adafruit.com/product/2693
**ADC**	Convertor analogic-digital (ADC MCP 3008) with eight channels. It allows to integrate the analogic sensors into the development board on the digital pins.	1	https://www.adafruit.com/product/856
**SNS-MQ135**	Analog sensor for gases (CO_2_, NH_4_, ethanol) detection.	1	https://www.olimex.com/Products/Components/Sensors/Gas/SNS-MQ135/
**MQ9**	Analogic sensor for gas (carbon monoxide (CO)) detection.	1	https://www.hotmcu.com/analog-cocombustible-gas-sensormq9-p-163.html
**SYH-2R**	Analog sensor for humidity level measurements.	1	https://www.tme.eu/en/details/syh-2rc/humidity-sensors/samyoung/syh-2r/
**Adafruit BMP 280**	Digital sensor for temperature, pressure, and altitude measurements.	1	https://www.adafruit.com/product/2651
**LEDs**	Simple colored LEDs (light emitting diodes) for flagging the station status (online/offline/recording events).	3	https://www.adafruit.com/product/1938
**Connection wires female-male**	Thin and small connection wires female-male (10 cm).	20	https://www.adafruit.com/product/3633
**Mini breadboard**	Breadboard (46 mm × 35 mm)	1	https://www.adafruit.com/product/65

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
