# Peer review of "IoT Solution for Smart Cities’ Pollution Monitoring and the Security Challenges"

_sensors, 2019, doi:10.3390/s19153401_

Round 1

Reviewer 1 Report

The authors describe the implantation of an IoT platform for monitoring multiple polluting gases in the city of Bucharest. The article is interesting although it written as tutorial. The authors have spent substantial time and resources to setup the platform but the paper is missing some critical focus point. For example, the authors identify environmental pollution as a major problem meanwhile the proposed platform monitors the air pollution but other that does not propose any radical solution for the problem. Second for sensors, they measure only air pollutants and report them back in a database (no critical control functions for infrastructure) why do you need extra security for something like that, perhaps the authors should concern more about the reliability of the measurements since they are using low cost non-calibrated sensors rather than the secure transmission of data. I recommend the authors to make improvements before publishing.

Author Response

Hello,

Thank you for your review. We have taken into account all of your comments, as it follows:

We tried to reshape and modify the paper in order to be less a tutorial and closer to a scientific study - please see section 4.2. We have also deleted some figures and added some case studies into the conclusions. 

This is a technical solution for monitoring the air pollution, mainly caused by traffic. The radical approach would be to integrate it with traffic management systems  – cameras monitoring and traffic lights – in order to fully make use of its power and benefits. This solution could reduce the pollution degree by dynamically offering alternate routes for the vehicles or even enforcing re-routing when pollution thresholds are reached.

The solution uses PoC hardware devices so there may be small calibration issues - for real, powerful devices these issues would be eliminated. 

The security is important, in order to avoid indirect attacks on the data collection process. The system also needs to be resilient to the malicious threats – e.g. cyber terrorism - that could represent high risks once the integration with any traffic control system is made

Please see our next version of the paper and we are looking forward for your feedback!

Best regards,

Cristian Toma

Reviewer 2 Report

I believe it is a very good paper that aims to present an Internet of Things solution with enhanced security for the sensors and gateway for monitoring the pollution within smart cities. The proposed Internet of Things (IoT) Solution emphasis the architecture of the components, the data flow implementation details and the edge connectivity to the IoT Clouds. Also, best practices and guidance for IoT infrastructure architects and developers are pointed by the authors of this paper.

Even though the technical information gives clear enough, they should have provided some more information about how they provide it and also more detailed theoretical information.  

Suggestions which would improve the quality of the paper but are not essential for publication:

1.      The paper has a number of English grammatical errors. Please correct the paper by asking a native English speaker to check it.

2.      Abstract need to write in better way and format. The major contributions of this work also need to be added in the Abstract.

3.      Recent Research: More recent references need to be added and discussed in the related work and background information about existing approaches in the context of proposed work addressed in this paper. Although the authors have conducted a thorough literature review, some very important references are still missing. The following references should be included in the revision:

ü  C. Stergiou, K. E. Psannis, A. P. Plageras, Y. Ishibashi, B.-G. Kim, “Algorithms for efficient digital media transmission over IoT and cloud networking”, Journal of Multimedia Information System, vol. 5, no. 1, pp. 1-10, March 2018.

ü  Vaesileios Memos, Kostas E. Psannis*, Yutaka Ishibashi, Byung-Gyu Kim, Brij Gupta, An Efficient Algorithm for Media-based Surveillance System (EAMSuS) in IoT Smart City Framework, Elsevier,  Future Generation Computer Systems, 2017.

ü  A. P. Plageras, K. E. Psannis, C. Stergiou, H. Wang, B. B. Gupta, “Efficient IoT-based sensor BIG Data collection-processing and analysis in Smart Buildings”, Future Generation Computer Systems, vol. 82, pp. 349-357, May 2018.

ü  C. Stergiou, K. E. Psannis, A. P. Plageras, Y. Ishibashi, B.-G. Kim, “Algorithms for efficient digital media transmission over IoT and cloud networking”, Journal of Multimedia Information System, vol. 5, no. 1, pp. 27-34, March 2018.

ü  C. Stergiou, K. E. Psannis, B. B. Gupta, “Advanced Media-based Smart Big Data on Intelligent Cloud Systems”, IEEE Transaction on Sustainable Computing, in Press, 2018.

4.      Emphasize the merits of the proposed approach in separate section.

5.      Present main contributions in Introduction section.

6.      Give a more detailed description about the figures.

7.      Conclusion section needs to be written in a better way. A more precise conclusion is needed. Add some future work in Conclusion section. It would be good as a future work to add something as case study.

Author Response

Hello,

Thank you for the feedback! We have addressed your comments, as it follows:

We have tried to obtain help for the English from the paper from different collaborators.

The abstract with contribution has been updated.

More recent references has been added in this paper - please see reference [10], [15] and [31].

Section 4.2 emphases the proposed approach merits.

The Introduction section has been modified with main contributions.

More detailed description has been added to several figures.

Conclusions updates and case studies have been added.

Please see the next version of the proposed scientific paper and we are looking forward for your feedback!

Best regards,

Cristian Toma

Round 2

Reviewer 1 Report

The authors addressed sufficient my previous comments, hence I believe it should be published. Just a comment to authors for future work they should use, calibrated high quality sensors for reporting directly to World Health Organization in order to maintain the quality of data high and reliable equivalent to their research reputation.

Author Response

Dear reviewer,

Thank you for the second feedback! We have addressed your second comments, as it follows:

In the Introduction section, we have modified the sentence:"For the proof of the concept and the prototyping the authors have used the development boards such as Nitrogen iMX 6 or Raspberry Pi, but for the final IoT solution, proper calibrated high quality sensors and industrial IoT gateway devices such as HMS Netbiter or equivalent equipment are targeted."

In the conclusion section, we have inserted the following sentence: "The authors are also targeting to use calibrated high quality sensors and interface implementation for reporting data directly to World Health Organization, in order to maintain the quality of data for the future work."

These are highlighting your second recommendation.

Best regards,

Cristian Toma

Reviewer 2 Report

I believe it is a very good paper that aims to present an Internet of Things solution with enhanced security for the sensors and gateway for monitoring the pollution within smart cities. The proposed Internet of Things (IoT) Solution emphasis the architecture of the components, the data flow implementation details and the edge connectivity to the IoT Clouds. Also, best practices and guidance for IoT infrastructure architects and developers are pointed by the authors of this paper.

Even though the technical information gives clear enough, they should have provided some more information about how they provide it and also more detailed theoretical information.  

Suggestions which would improve the quality of the paper but are not essential for publication:

1.      The paper has a number of English grammatical errors. Please correct the paper by asking a native English speaker to check it.

2.      Abstract need to write in better way and format. The major contributions of this work also need to be added in the Abstract.

3.      Recent Research: More recent references need to be added and discussed in the related work and background information about existing approaches in the context of proposed work addressed in this paper. Although the authors have conducted a thorough literature review, some very important references are still missing. The following references should be included in the revision:

ü  C. Stergiou, K. E. Psannis, A. P. Plageras, Y. Ishibashi, B.-G. Kim, “Algorithms for efficient digital media transmission over IoT and cloud networking”, Journal of Multimedia Information System, vol. 5, no. 1, pp. 1-10, March 2018.

ü  Vaesileios Memos, Kostas E. Psannis*, Yutaka Ishibashi, Byung-Gyu Kim, Brij Gupta, An Efficient Algorithm for Media-based Surveillance System (EAMSuS) in IoT Smart City Framework, Elsevier,  Future Generation Computer Systems, 2017.

ü  A. P. Plageras, K. E. Psannis, C. Stergiou, H. Wang, B. B. Gupta, “Efficient IoT-based sensor BIG Data collection-processing and analysis in Smart Buildings”, Future Generation Computer Systems, vol. 82, pp. 349-357, May 2018.

ü  C. Stergiou, K. E. Psannis, A. P. Plageras, Y. Ishibashi, B.-G. Kim, “Algorithms for efficient digital media transmission over IoT and cloud networking”, Journal of Multimedia Information System, vol. 5, no. 1, pp. 27-34, March 2018.

ü  C. Stergiou, K. E. Psannis, B. B. Gupta, “Advanced Media-based Smart Big Data on Intelligent Cloud Systems”, IEEE Transaction on Sustainable Computing, in Press, 2018.

4.      Emphasize the merits of the proposed approach in separate section.

5.      Present main contributions in Introduction section.

6.      Give a more detailed description about the figures.

Conclusion section needs to be written in a better way. A more precise conclusion is needed. Add some future work in Conclusion section. It would be good as a future work to add something as case study.

Author Response

Dear reviewer,

Thank you for the second feedback! We have addressed your second comments, as it follows:

We have tried to obtain help for the English from the paper from different collaborators.

The abstract with contribution has been updated with the major authors contributions. 

More recent references has been added in this paper - please see reference [10], [15] and [31].

The section 4.2 emphases the proposed approach merits.

The Introduction section has been modified with main contributions.

More detailed descriptions have been added to several figures.

Conclusions updates and case studies have been added.

We have provided more information about the solution in order to meet the technical and theoretical requests.

Please see this version of the proposed scientific paper and we are looking forward for your feedback!

Best regards,

Cristian Toma
